# The Difference in Clinical Behavior of Gene Fusions Involving *RET/PTC* Fusions and *THADA/IGF2BP3* Fusions in Thyroid Nodules

**DOI:** 10.3390/cancers15133394

**Published:** 2023-06-28

**Authors:** George Tali, Alexandra E. Payne, Thomas J. Hudson, Sabrina Daniela da Silva, Marc Pusztaszeri, Michael Tamilia, Véronique-Isabelle Forest

**Affiliations:** 1Faculty of Medicine, McGill University, Montreal, QC H3A 0G4, Canada; 2Health Science Program, Marianopolis College, Westmount, QC H3Y 1X9, Canada; 3Department of Otolaryngology-Head and Neck Surgery, McGill University, Montreal, QC H3A 0G4, Canada; 4Department of Pathology, Jewish General Hospital, Montreal, QC H3T 1E2, Canada; 5Division of Endocrinology, Jewish General Hospital, Montreal, QC H3T 1E2, Canada; 6Department of Otolaryngology-Head and Neck Surgery, Jewish General Hospital, Montreal, QC H3T 1E2, Canada

**Keywords:** thyroid, endocrine, cancer, RET/PTC, THADA/IGF2BP3, thyroidectomy, molecular panel, genetic translocation, Bethesda score

## Abstract

**Simple Summary:**

Around 25% of patients who undergo an ultrasound-guided thyroid biopsy end up with an indeterminate result based on cytology. This has propelled the use of other modalities, such as molecular testing, to further stratify these patients. The aim of our retrospective study was to report and compare two genetic mutations in our patient population. These mutations are RET/PTC and THADA/IGF2BP3 translocations, which have been hypothesized as oncogenic events in thyroid neoplasms. We confirm that our patient population exhibited these mutations, and all underwent a final histopathology analysis where surgery was the preferred treatment modality. We also report that the RET/PTC fusion exhibited more aggressive features than the THADA/IGF2BP3 fusion and was more likely to need post-surgical treatment.

**Abstract:**

Background: Molecular testing has been used as an adjunct to morphological evaluation in the workup of thyroid nodules. This study investigated the impact of two gene fusions, *RET/PTC* and *THADA/IGF2BP3*, that have been described as oncogenic events in thyroid neoplasms. Methods: We performed a retrospective, single-centered study at a McGill University teaching hospital in Montreal, Canada, from January 2016 to August 2021. We included patients who underwent surgery for thyroid nodules that pre-operatively underwent molecular testing showing either *RET/PTC* or *THADA/IGF2BP3* gene fusion. Results: This study included 697 consecutive operated thyroid nodules assessed using molecular testing, of which five had the *RET/PTC* fusion and seven had the *THADA/IGF2BP3* fusion. Of the five nodules in the *RET/PTC* group, 100% were malignant and presented as Bethesda V/VI. Eighty percent (4/5) were found to have lymph node metastasis. Twenty percent (1/5) had extrathyroidal extensions. Sixty percent (3/5) were a diffuse sclerosing variant of papillary thyroid carcinoma, and the rest were the classical variant. Of the seven *THADA/IGF2BP3* nodules, all presented as Bethesda III/IV and 71.4% (5/7) were malignant based on the final pathology analysis, and 28.6% (2/7) were NIFTP. All the *THADA/IGF2BP3* fusion malignancies were a follicular variant of papillary thyroid carcinoma. None had lymph node metastasis or displayed extrathyroidal extensions. Conclusions: *RET/PTC* nodules presented as Bethesda V/VI and potentially had more aggressive features, whereas *THADA/IGF2BP3* nodules presented as Bethesda III/IV and had more indolent behavior. This understanding may allow clinicians to develop more targeted treatment plans, such as the extent of surgery and adjuvant radioactive iodine treatment.

## 1. Introduction

### Thyroid Cards

Cancer is the most common endocrine malignancy. Its incidence has been steadily increasing over the recent decades and can be attributed to the advancements in diagnostic modalities that have facilitated the detection of thyroid nodules earlier in the disease process [1,2]. The gold standard for the workup of suspicious thyroid lesions is the ultrasound-guided fine needle aspiration (USFNA) [3,4]. Although cytopathology classifies the majority of biopsies as benign or malignant, 20–25% of nodules are still classified as indeterminate, i.e., Bethesda III (atypia of undetermined significance) and Bethesda IV (follicular neoplasm or suspicious for follicular neoplasm) [5,6]. This has propelled the field of molecular testing as a concurrent modality to further diagnose and prognosticate thyroid nodules.

The latest iteration of the American Thyroid Association (ATA) guidelines acknowledges the use of molecular testing as an adjunct to USFNA in the workup of cytologically indeterminate nodules, provided patients are educated regarding the benefits and limitations of genetic testing. The ATA guidelines also discuss specific mutations in terms of their aggressive behavior to predict risk of recurrence and guide post-operative management [7]. It is crucial to identify the molecular markers not only to distinguish between benign and malignant tumors but also to predict aggressive phenotypes, prognosis, recurrence, and efficacy of treatment, including potential novel therapeutic targets [8].

The Bethesda scoring system is based on cytopathologic features of FNA biopsy. Score II represents potential benign thyroid nodule where monitoring using ultrasound and/or blood testing is recommended, and chance of significant growth remains low enough to not warrant surgical intervention [9]. Intermediate Bethesda scores (III/IV) represent a follicular lesion/atypia of undetermined significance. This entity of indeterminate thyroid nodules often presents a dogma to clinicians. Much of the literature has focused on this problematic entity, with the probability of having a malignant pathology often significantly fluctuating between different institutions and regions [10,11,12]. Bethesda score V includes lesions with features suspicious for, but not definitive for, a malignant thyroid pathology, and Bethesda VI lesions are presumed malignant and referred for surgical management [5].

There are several types of thyroid cancer, which differ in their presentation, aggressiveness, survival rates, and treatments. The most common types are papillary thyroid cancer (PTC) and follicular thyroid cancer (FTC), which together make roughly 90% of all cases [13]. Similarly, there are several variants of PTC that also differ in their prevalence, aggressiveness, and risk factors. One of the most aggressive variants that is relevant to this study is diffuse sclerosing variant (DSV), which is most frequently observed in younger patients [14]. It is very aggressive with a high risk of extrathyroidal extension, cervical lymph node metastasis, and recurrence [14,15].

A subtype of FTC is Hürthle cell carcinoma (HCC), which is more aggressive than FTC and makes up around 3% of all cases. Another subtype of FTC is noninvasive follicular thyroid neoplasm with papillary-like nuclear features (NIFTP), which was previously classified as a type of PTC.

Molecular studies have identified many oncogenic drivers in thyroid cancer. Amongst known drivers, 75% are described as chromosomal point mutations, 15% show nonoverlapping chromosomal rearrangements that generally lead to the activation of tyrosine kinases, and 10% where no driver is identified [16,17,18]. In addition to their diagnostic capabilities, molecular panels may also aid in the future in providing targeted therapies that can inhibit the activity of an aberrant driver pathway in thyroid tumorigenesis [19,20]. Indeed, the advent of molecular therapy is now considered gold standard in the treatment of certain anaplastic thyroid cancer (ATC), whereby the determination of the mutation status of BRAF gene is recommended in the ATA guidelines [21]. Specifically, treatment with dabrafenib has shown some efficacy in BRAF-mutated thyroid cancer, especially when combined with mitogen-activated protein kinase kinase (MEK) inhibitors, such as trametinib [22].

One gene rearrangement of interest is rearranged during transfection (RET)*. RET* is a proto-oncogene that has been implicated in PTC and medullary thyroid carcinoma (MTC) [23]. The RET gene is located in the long arm of chromosome 10 and encodes a cell membrane tyrosine kinase whose activation stimulates mitogen-activated protein kinase (MAPK) and PI3K, which promote downstream cellular proliferation [24,25]. *RET/PTC* refers to the fusion of *RET* with one of the different heterologous genes to create a series of chimeric oncogenes [26]. This rearrangement, in turn, produces a protein that activates MAPK pathway [27]. It has been reported in up to 20–40% of adult sporadic papillary carcinomas; however, its prevalence is highly variable amongst studies [26]. The presence of *RET/PTC* translocation has also been implicated in tumor multifocality [28]. *RET/PTC* fusions have been shown to exist concomitantly with other mutations, namely *BRAF(V600E)* in malignant PTC [29]. RET/PTC fusions are also different from RET point mutations, which are single changes in the DNA sequence of the RET gene. These point mutations can be either inherited or acquired and have been implicated in MTC [27].

Another gene rearrangement of interest is referred to as thyroid adenoma associated (*THADA*) gene fusion, which is the second most common chromosomal rearrangement in thyroid cancer [26]. It is hypothesized to have a prevalence of around 5% of thyroid cancer that lack any other identifiable driver mutation [30]. Recent studies by Panebianco et al. have elucidated the mechanism by which *THADA*-gene fusion leads to increased cellular growth, migration, and invasion. Translocations between chromosomal bands 2p21 (*THADA*) and 7p15 lead to overexpression of IGF2, thereby activating downstream MAPK and PI3K signaling pathways [30,31,32,33]. It can also occur in other cancer types, such as breast, ovarian, lung, and colorectal cancers [30].

To date, several studies have assessed the prevalence and clinicopathological characteristics of these two genetic rearrangements. *RET/PTC* fusion prevalence has been reported in few studies with large variance [34]. *THADA/IGF2BP3* fusion has been recently studied by Morariu et al. where a 2% prevalence was reported amongst cytologically indeterminate thyroid nodules [35]. The aim of the study was to establish the prevalence of *RET/PTC* and *THADA/IGF2BP3* in our patient population as well as to study the histopathology, management, and surgical outcomes. Such insight will assist in the pre-operative/perioperative decision-making for patients with one of these molecular alterations.

## 2. Materials and Methods

### 2.1. Study Design

A retrospective chart review was performed on patients who were worked up for thyroid nodules from January 2016 to August 2021. All patients were treated at the Jewish General Hospital (JGH), Faculty of Medicine, McGill University in Montreal, Quebec. The study included patients that had undergone pre-operative molecular testing using one of the following commercially available tests: Afirma GSC, Thyroseq V3, or ThyGeNEXT/ThyraMIR. They later underwent surgery.

Patients were excluded if surgical pathology results were not available. None of study subjects had more than one driver mutation reported using molecular testing. Patients who had multiple nodules that were biopsied, with different cytopathology classifications, the higher Bethesda score was used to calculate the prevalence of different Bethesda scores and corresponding malignancy based on the final pathology analysis. Baseline characteristics, such as demographics, cytopathology, surgical pathology, and results of molecular/genetic findings, treatment, and outcome were collected for analysis (Table 1). In case of malignancy, the specific type (papillary thyroid carcinoma, follicular carcinoma, Hürthle cell carcinoma, and poorly differentiated thyroid carcinoma) and the variant (classical, follicular, oncocytic, tall cell, columnar cell, solid, and hobnail) were recorded (Table 1). Operating room reports were examined in detail for any complications or findings that were not seen on prior evaluation.

### 2.2. Data Collection

Ethics approval for the purpose of this study was granted by the JGH Research Ethics Committee (2021-2617). Using patient identifiers in logs, the charts were accessed. This included baseline characteristics, such as age and USFNA results as per the Bethesda reporting system. Medical documents, such as reports, clinician notes, labs, and imaging of the patients were accessed.

### 2.3. Statistical Analysis

Statistical analysis was performed using SPSS to compare patient demographics and tumor characteristics between the genetic aberrations under study. Chi-squared analysis was used to calculate confidence intervals. Continuous variables, including Bethesda score, were compared using independent sample *t*-test or ANOVA. Statistical significance was set as *p* < 0.05.

## 3. Results

### 3.1. Prevalence

During the study period, 697 consecutive thyroid nodules were analyzed using molecular analysis in the pre-operative workup. Out of these thyroid nodules, 377 were Bethesda III or IV on USFNA and 312 were Bethesda V or VI on USFNA (see Figure 1). Amongst Bethesda III nodules, 105 out of a total of 165 (64%) were malignant or non-invasive follicular thyroid neoplasm with NIFTP in the final histopathology analysis; amongst Bethesda IV nodules, 159 out of a total of 212 (73%) were malignant or NIFTP in the final histopathology analysis.

All USFNA biopsies that were *RET/PTC* fusion positive were reported as Bethesda V or VI. All USFNA biopsies that were *THADA/IGF2BP3* fusion positive were reported as Bethesda III or IV. Out of the 312 nodules with Bethesda V or VI on USFNA, five (1.60%) were positive for a *RET/PTC* fusion. Out of the 377 nodules with Bethesda III or IV on USFNA, seven (1.86%) were positive for a *THADA/IGF2BP3* fusion. Overall, all nodules with *RET/PTC* or *THADA/IGF2BP3* fusions were either malignant or NIFTP in the final histopathology analysis.

### 3.2. Baseline Characteristics

Baseline information was calculated for all 12 patients. Tumor size was reported as per the long-axis measurement in centimeters in the final pathology reports. The baseline characteristics are summarized in Table 1. No statistically significant differences were detected with regards to age, gender, or final pathologic tumor size between the *RET/PTC* and *THADA/IGF2BP3* fusion cohorts (*p* = 0.064, *p* = 0.154, and *p* = 0.104, respectively). *RET/PTC* fusion translocations presented with a higher Bethesda Score Distribution than *THADA/IGF2BP3* fusion translocations (*p* < 0.01). All patients underwent a sentinel lymph node (LN) biopsy intraoperatively in addition to a limited central neck dissection. If the sentinel lymph node is positive, then a more extensive central neck dissection is performed by the surgeon.

### 3.3. Cancer Type and Correlation with Clinicopathological Characteristics

All the *RET/PTC* fusion cases (5/5) presented as Bethesda V/VI. They were all positive for malignancy in the final histopathology analysis: 60% (3/5) were diffuse sclerosing variant (DSV) and the rest were classical variant of PTC. All DSV cases were positive for lymphovascular invasion, while the nodules with classical variant were all negative for lymphovascular invasion. None of the nodules were positive for perineural invasion. Only one case (1/5) was positive for extrathyroidal extension (minimal). The final pathology analysis reported 80% (4/5) of those patients had LN metastasis, with one of these cases presenting a lymph node with extranodal spread (ENS). All patients underwent one sentinel LN biopsy, which if positive, the surgeon proceeded to conduct a central compartment LN biopsy (see Table 2). All positive LNs had the shortest dimension of at least 0.2 cm.

Amongst the *THADA/IGF2BP3* fusion group, all (7/7) presented as Bethesda III or IV. Furthermore, 71.4% (5/7) were malignant in the final pathology analysis, and 28.6% (2/7) were NIFTP [36]. Amongst malignant nodules, all presented as follicular variant of PTC. None of the nodules was positive for extrathyroidal extension. Only one nodule that presented as a follicular PTC was positive for focal lymphovascular invasion. None of the nodules had perineural invasion. In the final pathology analysis, none of the patients had LN metastasis or extranodal spread (see Table 2).

### 3.4. Patient Management

Of the five *RET/PTC* fusion cases, four underwent total thyroidectomy and one had undergone a hemi/subtotal thyroidectomy. The patient who underwent a hemi/subtotal thyroidectomy was the only patient whose intraoperative sentinel LN biopsy was negative for malignancy. Of the seven *THADA/IGF2BP3* fusion cases, all were treated with hemi/subtotal thyroidectomy. None of these patients had a positive intraoperative sentinel LN biopsy. None underwent a completion thyroidectomy (see Table 2).

### 3.5. Patient Follow-Up

Five patients with *THADA/IGF2BP3* fusion nodules had available data for follow-up. Four of these patients had malignant nodules and one with NIFTP. Mean follow-up period was 18 months (range 1–33 months). No recurrences were identified during that period.

Five patients with *RET/PTC* fusion nodules had follow-up data, where the mean follow-up period was 20.6 months (range 1–28 months). Three patients had thyroid scans consistent with remnant thyroid tissue and needed to undergo additional treatment. The patients who had positive post-surgical thyroid scans were the same patients that had diffuse sclerosing variant malignancies in surgical pathology.

## 4. Discussion

Over the last decade, the advent of molecular testing has been rapidly expanding in the pre-operative workup of thyroid nodules. This single-centered study was conducted in Montreal, Quebec, where molecular testing was offered to patients on a case-per-case and voluntary basis. Molecular testing panels have generally branded themselves as a strong rule-in or rule-out, reducing the need for diagnostic surgery in thyroid nodules with intermediate Bethesda scores [7].

The identification of various molecular markers during thyroid cell transformation and tumor progression is a critical step in understanding the underlying pathogenetic factors that can guide clinical management of thyroid cancer. Several genetic alterations have been reported thus far, which include passenger mutations and genetic drivers, such as *BRAF*, *p53*, *NRAS*, *KRAS*, and *HRAS* [37]. However, the *RET/PTC* and *THADA/IGF2BP3* fusions are less well-studied as they comprise a small percentage of known genetic alterations. This retrospective study aimed to evaluate the clinical impact of these two genetic rearrangements as well as shed light on the pragmatic application of molecular testing in the workup and management of thyroid nodules.

Our retrospective study establishes an estimate of the prevalence of both mutations in our patient population, within their respective Bethesda buckets. All nodules harboring a *RET/PTC* fusion were Bethesda V/VI in the cytology analysis. In this context, the extent of surgery was based on available clinical information, patient preference as well as intraoperative findings. Most of the patients with a *RET*/PTC fusion in this study had lymph node metastases and/or extrathyroidal extension. Moreover, more than half the cases were a diffuse sclerosing variant. The knowledge that a thyroid nodule has a *RET/PTC* fusion with a high likelihood of being aggressive will guide thyroid specialists to making a more informed decision with respect to the extent of surgery, including the decision to perform a central neck dissection.

On the other hand, all cases of nodules with a *THADA/IGF2BP3* fusion presented as indeterminate cytology in the USFNA assessment. All the nodules required surgery as the final pathology was malignant or NIFTP. While most of these nodules (5/7) were malignant in the final surgical pathology, none were considered as aggressive or at intermediate or higher risk for recurrence. As a result, a more conservative and limited surgical approach might be warranted in nodules with a *THADA/IGF2BP3* fusion. These findings are in line with other reported findings by Panebianco et al. and Morariu et al.; however, the prevalence in the *THADA/IGF2BP3* fusion nodules was lower in our patient population [30,35].

The findings of this study highlight the potential for optimizing patient care and management decisions based on molecular testing results. Prior to the release of the American Thyroid Association Guidelines in 2015 for Differentiated Thyroid Cancer, many thyroid malignancies were treated with total thyroidectomies and radioactive iodine. Following the recommendations in these guidelines, practice regimens changed, and less extensive surgeries (hemi-thyroidectomy/lobectomy) gained popularity. Whether to perform a total thyroidectomy or hemi-thyroidectomy for patients with thyroid malignancies became a management dilemma. Performing a hemi-thyroidectomy can lead to a completion thyroidectomy when the tumor is found to be aggressive in the final pathology analysis. As a result, the patient requires a second surgery. This gives rise to increased costs and resource allocation associated with this additional intervention. Additionally, the need for the patient to take out more time from their usual schedules (e.g., work, etc.) would increase. Accordingly, when a total thyroidectomy is performed for a thyroid malignancy that is not considered to be aggressive, the extent of this surgery may be considered as overtreatment. Moreover, total thyroidectomy is a longer intervention with more associated complications. In addition, patients are dependent on lifelong levothyroxine supplementation.

This study clearly demonstrates the value of molecular testing to help optimize the extent of surgery and avoid any unnecessary total thyroidectomies. Indeed, a patient with a *THADA/IGF2BP3* fusion will likely benefit from a limited surgery, such as a hemi-thyroidectomy or lobectomy, as patients in this group either had low-grade malignancies or NIFTP tumors. These findings indicate that *THADA/IGF2BP3* is generally a non-aggressive mutation. This knowledge can be implemented in patient care and the way tumors with these mutations are approached. Adversely, patients with a *RET/PTC* fusion had an 80% likelihood of exhibiting an aggressive disease, and thus, a total thyroidectomy is potentially the preferred surgery for this group of patients. Moreover, given that the likelihood of lymph node metastasis is elevated in this group, consideration for a central compartment neck dissection should be strongly considered.

There were several limitations in this study. The sample size of cases with *RET/PTC* or *THADA/IGF2BP3* fusions is limited, restricting the power of our statistical inferences. The follow-up periods were limited, and the recurrence rates for both fusions are unknown. The study is also limited by the inherit weaknesses of retrospective studies. In addition, the surgeons and pathologists were not blinded to the results of the molecular test. As a result, the surgeon may have been more likely to perform a more comprehensive exploration of the central compartment for abnormal lymph nodes in the *RET/PTC* group than for patients found to have a *THADA/IGF2BP3* fusion. When the mutation is a *THADA/IGF2BP3* fusion, the surgeon might be less inclined to perform this lymph node exploration because they are aware that the likelihood of lymph node metastases is significantly less. The pathologist may have been biased towards the results of the molecular testing.

## 5. Conclusions

In this retrospective chart review analyzing *RET/PTC* and *THADA/IGF2BP3* fusions, all were found to be malignant or NIFTP. The *RET/PTC* fusion group had more aggressive tendencies (extrathyroidal extension, lymph node metastasis, and diffuse sclerosing variant) and was classified as intermediate risk of recurrence as per the 2015 American Thyroid Association Guidelines for Differentiated Thyroid Cancer. Whereas the *THADA/IGF2BP3* fusion nodules had a more indolent behavior, classified as either low-risk malignancy or NIFTP in the final pathology analysis. This understanding of genetic drivers may allow clinicians to develop more targeted treatment plans and supports the notion that molecular testing has significant clinical value in the pre-operative workup of cytologically indeterminate nodules. This study confirms that further research assessing *RET/PTC* and *THADA/IGF2BP3* rearrangements in suspicious thyroid nodules is necessary to improve patient management, particularly for deciding if a targeted therapy plan can be developed. In resource-limited healthcare systems, performing the optimal surgery the first time is beneficial for the patient, the system, and the thyroid specialist. The knowledge of how to handle specific thyroid mutations, such as *RET/PTC* and *THADA/IGF2BP3*, can aid surgeons in making the most informed choices and maximize the efficiency of the surgical process. Similarly, understanding how to treat certain mutations can prevent their reoccurrence, which is advantageous for the patient and the physician. This study highlights the value of molecular testing not only as a rule out test to avoid unnecessary surgery but also as a tool to help determine the extent of surgery and need for adjuvant treatment.

## Figures and Tables

**Figure 1 cancers-15-03394-f001:**
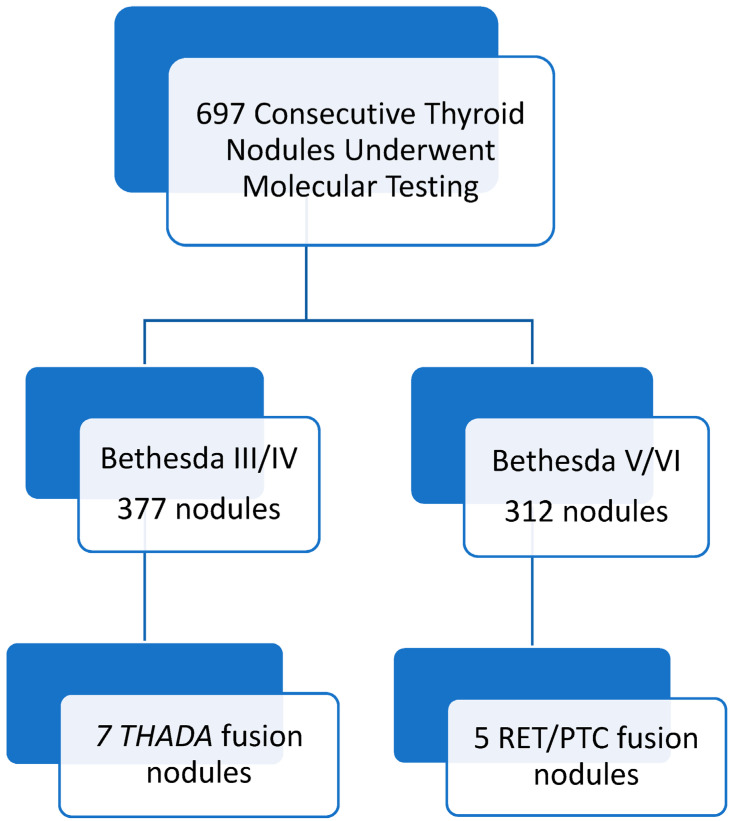
Summary of log search results pertaining to both genetic translocations of interest.

**Table 1 cancers-15-03394-t001:** Baseline Characteristics of Patients. FNA: fine needle aspiration.

Baseline Characteristics								
Mutation	n	Mean Age (Years)	Gender (% Female)	FNA Bethesda Score Distribution				Final Pathology Size (cm)
				III	IV	V	VI	
*THADA/IGF2BP3*	7	51.7	71%	4	3	0	0	2.06 (1–3.6)
*RET/PTC*	5	38.7	100%	0	0	1	4	1.15 (0.7–1.8)

**Table 2 cancers-15-03394-t002:** Baseline characteristics, procedure undergone, and final pathological history. NIFTP: noninvasive follicular thyroid neoplasm with papillary-like nuclear features; DSV: diffuse sclerosing variant.

*THADA/IGF2BP3* Fusion	Patient	Age (Years)	Gender	Bethesda	Pathology Size (cm)	Procedure	TNM	Histology	Variant	Lymph Node Metastasis	Positive LN/LN Examined	Positive Central LN/Central LN Examined	Extranodal Spread	Extrathyroidal Extension	Lymphovascular Invasion	Perineural Invasion
	1	64	Male	4	1.3	Hemi/subtotal thyroidectomy	T1bN0a	Papillary Ca	Follicular	Negative	0/2	-	No	No	No	No
	2	53	Female	3	1.9	Hemi/subtotal thyroidectomy	T1bN0a	Papillary Ca	Follicular	Negative	0/2	-	No	No	No	No
	3	62	Female	4	1.2	Hemi/subtotal thyroidectomy	T1bN0a	Papillary Ca	Follicular	Negative	0/1	-	No	No	No	No
	4	48	Female	3	3.4	Hemi/subtotal thyroidectomy	T2N0a	Papillary Ca	Follcicular	Negative	0/1	-	No	Yes (focal)	No	No
	5	47	Male	4	1.0	Hemi/subtotal thyroidectomy	-	NIFTP	-	Negative	0/6	0/2	No	No	No	No
	6	47	Female	3	2.0	Hemi/subtotal thyroidectomy	T1bN0a	Papillary Ca	Follicular	Negative	0/1	-	No	No	No	No
	7	41	Female	3	3.6	Hemi/subtotal thyroidectomy	-	NIFTP	-	Negative	0/1	-	No	No	No	No
*RET/PTC* fusion																
	1	46	Female	6	1.8	Total Thyroidectomy	T1bN1a	Papillary Ca	Classical	Positive	1/3	0/1	No	No	No	No
	2	38	Female	6	0.7	Total Thyroidectomy	T1bN1a	Papillary Ca	DSV	Positive	4/5	3/4	No	Yes (minimal)	Yes (extensive)	No
	3	31	Female	6	0.9	Total Thyroidectomy	T1bN1a	Papillary Ca	DSV	Positive	1/2	0/1	No	No	Yes (extensive)	No
	4	34	Female	6	1.2	Total Thyroidectomy	T1bN1a	Papillary Ca	DSV	Positive	4/4	2/2	Yes	No	Yes	No
	5	57	Female	5	1.2	Hemi/subtotal Thyroidectomy	T1bN0a	Papillary Ca	Classical	Negative	0/1	-	No	No	No	No

## Data Availability

The data presented in this study are available in this article.

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
