# Peer review of "The Difference in Clinical Behavior of Gene Fusions Involving RET/PTC Fusions and THADA/IGF2BP3 Fusions in Thyroid Nodules"

_cancers, 2023, doi:10.3390/cancers15133394_

Round 1

Reviewer 1 Report

The authors evaluate the clinical characteristics of gene fusions in differentiated thyroid cancer, particularly on RET/PTC and THADA/IGF2BP3 fusions. The authors identify distinct Bethesda cytologic classifications, histopathologic features, as well as clinical courses in these two gene fusions, with RET/PTC presenting with a more aggressive disease. Overall, the manuscript is well-written. There are several limitations to this study, predominantly the small sample size of the population in question. Therefore, additional studies are required to confirm several of these findings. However, the authors have mentioned all the limitations.

One minor point is that the term USFNA has already been described in the ‘Introduction’ section, and it is expanded once again in the ‘Data Collection’ section, where only the abbreviation can be used.

Author Response

Dear Reviewer,

Thank you for taking the time to review our manuscript.

We have fixed the manuscript per your suggestions.

Sincerely,

George Tali

Reviewer 2 Report

Tali et al. to establish the prevalence of RET/PTC and THADA/IGF2BP3 in observed patient population, aiming to improve the pre-operative/perioperative decision-making in patients with one of these molecular alterations. The title, Intro, M and M, Stats, Results, Discussion and Conclusion sections are done as they do. The Refs are correct and appropriate. 

Ready for publishing. 

Author Response

Dear Reviewer,

Thank you for taking the time to review our manuscript.

Sincerely,

George Tali

Reviewer 3 Report

This is a very nice study assessing the effect of two important gene fusions on the risk of malignancy and the histological features of the tumors associated with these fusions. The study protocol was designed correctly, and the analysis was apporopriate. The findings are important and highlight the different outcomes with regard to the risk of malignancy when the RET/PTC vs. the THADA/IGF2BP3 gene fusions are found in thyroid nodules. RET/PTC appears to be a highly malignant gene fusion, while THADA/IGF2BP3 appears to be a rather benign one. 

Despite the significance of these findings in the present manuscript and the multiple merits it bears, it is limited by the presence of the extremely small number of cases in each subgroup. Therefore, I would recommend collecting additional data and adding some cases prior to resubmitting for publication. 

Author Response

Dear Reviewer,

Thank you for taking the time to review our manuscript.

We have fixed the manuscript per your suggestions.

Regrettably, we reviewed our department surgical logs in the hope of adding further cases to our sample. Since our initial submission, 2 more cases have presented (one of each genetic aberration), but both do not have final histopathological reports, and therefore cannot be included.

Sincerely,

George Tali

Reviewer 4 Report

The manuscript by Tali and collaborators analyse clinical impact of RET/PTC or THADA/IGF2BP3 translocations in thyroid nodules. The study is interesting and well-written. Unfortunately, it also has a few limitations, as outlined in the Discussion section (e.g., sample size). Suggested changes are listed below.

Major remarks

1. Please explain why the analysis focuses on only two rearrangements, i.e., RET/PTC fusions and THADA/IGF2BP3. Why e.g. PAX8/PPARγ rearrangement was omitted?

2. It seems to me that in the Introduction it is worth explaining what are the types of thyroid cancer, especially those that are mentioned later in the manuscript (e.g. MTC, PTC, NIFTP, DSV). It is also worth mentioning the new 2022 WHO Classification of Thyroid Neoplasms.

3. All acronyms should be explained at the first mention in the main body of the manuscript.  For example, “NIFTP” is explained at the third mention. Please correct this.

4. Introduction: The Authors states that “In addition to their diagnostic capabilities, molecular panels may also aid in the future in providing targeted therapies that can inhibit the activity of an aberrant driver pathway in thyroid tumorigenesis… .” It is worth noting that in the treatment of ATC (stages IVB and IVC), determination of the mutational status of the BRAF gene is already recommended in ATA guidelines (2021). Treatment with dabrafenib (BRAF inhibitor) plus trametinib (MEK inhibitor) of BRAF (V600E)-positive ATC patients induce prompt and impressive tumor regression. It is worth citing here a review paper that summarizes the progress in targeted therapy for thyroid cancer (e.g., Int J Mol Sci. 2021 Oct 31;22(21):11829).

Minor remarks

1. Page 2, line 10 from the bottom: Change “MAPK [22]” for “MAPK pathway [22]”.

2. Explain FNA acronym in Table 1.

The manuscript by Tali and collaborators analyse clinical impact of RET/PTC or THADA/IGF2BP3 translocations in thyroid nodules. The study is interesting and well-written. Unfortunately, it also has a few limitations, as outlined in the Discussion section (e.g., sample size). Suggested changes are listed below.

Major remarks

1. Please explain why the analysis focuses on only two rearrangements, i.e., RET/PTC fusions and THADA/IGF2BP3. Why e.g. PAX8/PPARγ rearrangement was omitted?

2. It seems to me that in the Introduction it is worth explaining what are the types of thyroid cancer, especially those that are mentioned later in the manuscript (e.g. MTC, PTC, NIFTP, DSV). It is also worth mentioning the new 2022 WHO Classification of Thyroid Neoplasms.

3. All acronyms should be explained at the first mention in the main body of the manuscript.  For example, “NIFTP” is explained at the third mention. Please correct this.

4. Introduction: The Authors states that “In addition to their diagnostic capabilities, molecular panels may also aid in the future in providing targeted therapies that can inhibit the activity of an aberrant driver pathway in thyroid tumorigenesis… .” It is worth noting that in the treatment of ATC (stages IVB and IVC), determination of the mutational status of the BRAF gene is already recommended in ATA guidelines (2021). Treatment with dabrafenib (BRAF inhibitor) plus trametinib (MEK inhibitor) of BRAF (V600E)-positive ATC patients induce prompt and impressive tumor regression. It is worth citing here a review paper that summarizes the progress in targeted therapy for thyroid cancer (e.g., Int J Mol Sci. 2021 Oct 31;22(21):11829).

Minor remarks

1. Page 2, line 10 from the bottom: Change “MAPK [22]” for “MAPK pathway [22]”.

2. Explain FNA acronym in Table 1.

Author Response

Dear Reviewer,

Thank you for taking the time to review our manuscript.

We have fixed the manuscript per your suggestions.

Most of the driver rearrangements in the literature are reported superficially. We wanted to dive deep on a select number of rearrangements. In our surgical logs, the cases presenting with PAX8/PPARγ are significantly less frequent (3 total) that the reported rearrangements, and given that are a single-centered study, it would have been difficult to draw conclusions.

Sincerely,

George Tali

Round 2

Reviewer 3 Report

The present work is excellent in its design, data analysis and data presentation. Similarly to our prior comment, I fear that the numbers are inadequate to support the results and the conclusions aextrapolated from them.

Reviewer 4 Report

The Authors addresses all my concerns, therefore I would like to recommend the manuscript for publication.

I have only a few minor editorial suggestions:

1. Page 5: Correct “formed. which if positive”.

2. Table 2 legend: Please explain “LN” and “Ca”.

3. Page 6: Change “(see Table 2)” for “(Table 2)”.

The Authors addresses all my concerns, therefore I would like to recommend the manuscript for publication.

I have only a few minor editorial suggestions:

1. Page 5: Correct “formed. which if positive”.

2. Table 2 legend: Please explain “LN” and “Ca”.

3. Page 6: Change “(see Table 2)” for “(Table 2)”.